# Nanoshaped Cerium Oxide with Nickel as a Non-Noble Metal Catalyst for CO_2_ Thermochemical Reactions

**DOI:** 10.3390/molecules28072926

**Published:** 2023-03-24

**Authors:** Jarosław Serafin, Jordi Llorca

**Affiliations:** Institute of Energy Technologies, Department of Chemical Engineering and Barcelona Research Center in Multiscale Science and Engineering, Polytechnic University of Catalonia, Eduard Maristany 16, EEBE, 08019 Barcelona, Spain

**Keywords:** cerium dioxide, ceria nanoshapes, Ni/CeO_2_, thermochemical reaction, CO_2_ splitting, catalyst

## Abstract

Four different nanoshapes of cerium dioxide have been prepared (polycrystals, rods, cubes, and octahedra) and have been decorated with different metals (Ru, Pd, Au, Pt, Cu, and Ni) by incipient wetness impregnation (IWI) and ball milling (BM) methods. After an initial analysis based on oxygen consumption from CO_2_ pulse chemisorption, Ni-like metal, and two forms of CeO_2_ cubes and rods were selected for further research. Catalysts were characterized using the Brunauer-Emmett-Teller formula (BET), X-ray spectroscopy (XRD), Raman spectroscopy, scanning electron microscopy (SEM), UV–visible spectrophotometry (UV-Vis), X-ray photoelectron spectroscopy (XPS), temperature programmed reduction (H_2_-TPR) and CO_2_ pulse chemisorption, and used to reduce of CO_2_ into CO (CO_2_ splitting). Adding metals to cerium dioxide enhanced the ability of CeO_2_ to release oxygen and concomitant reactivity toward the reduction of CO_2_. The effect of the metal precursor and concentration were evaluated. The highest CO_2_ splitting value was achieved for 2% Ni/CeO_2_-rods prepared by ball milling using Ni nitrate (412 µmol/g_cat_) and the H_2_ consumption (453.2 µmol/g_cat)_ confirms the good redox ability of this catalyst.

## 1. Introduction

Currently, one idea for reducing the concentration of CO_2_ in the atmosphere is its conversion into products with a high positive value, i.e., chemicals or fuels [1,2]. There is a possibility of thermochemical conversion of CO_2_ using solar energy [3]. Research on thermochemical cycles dates to the 1960s and focused primarily on developing materials for nuclear reactors. The importance of thermochemical cycles in producing synthetic fuels has significantly increased with the approved worldwide climate protocols [4]. One of the most effective thermochemical cycles is the two-step redox oxide pair system, which has shown great potential for synthetic solar fuel generation [5]. The principle of operation is based on the transition between the higher valence oxidized and lower valence reduced form of the oxide of a metal having multiple oxidation states [6]. In the first stage, there is a thermal reduction of the metal oxide through the release of oxygen due to an endothermic reaction. The reduced metal oxide is oxidized in the second stage by taking oxygen from water and/or CO_2_. As a result, there is a return to the original oxidation state. This causes the production of H_2_ and CO in reactions called water splitting (WS) and carbon dioxide splitting (CDS), respectively [7]. Cerium oxide is considered an excellent candidate for use in thermochemical cycles due to its well-known redox properties [8,9] as well as the ability to form superficial oxygen vacancies during reduction, which are easily reoxidized [10]. This ability is described by Equations (1) and (2):

High temperature reduction:(1)$${\mathrm{CeO}}_{2-\mathrm{Xox}}\overset{T_{\mathrm{red}}}{\to }{\mathrm{CeO}}_{2-\mathrm{Xred}}+\frac{\Delta X}{2}O_{2}$$
low temperature oxidation with H_2_O:(2a)$${\mathrm{CeO}}_{2-\mathrm{Xred}}+{\Delta X\;H}_{2}O\overset{T_{\mathrm{ox}}}{\to }{\mathrm{CeO}}_{2-\mathrm{Xox}}+{\Delta X\;H}_{2}$$
low temperature oxidation with CO_2_:(2b)$${\mathrm{CeO}}_{2-\mathrm{Xred}}+{\Delta X\;\mathrm{CO}}_{2}\overset{T_{\mathrm{ox}}}{\to }{\mathrm{CeO}}_{2-\mathrm{Xox}}+\Delta X\;\mathrm{CO}$$

Equation (1) describes the endothermic reduction step, which is intended to perform using solar energy. Equation (2) describes the exothermic oxidation where CeO_2-Xred_ is re-oxidized by H_2_O to generate H_2_ (Equation (2a)) or by CO_2_ to generate CO (Equation (2b)).

Cerium oxide has the ability to store and release oxygen easily, which is directly related to the ability of cerium to change the oxidation state between Ce^4+^ and Ce^3+^ and the concomitant formation of oxygen vacancies. There are ideally three characteristic main crystallographic planes for cerium dioxide: {100} (cubes), {110} (rods), and {111} (octahedra). Figure 1 shows a scheme of the different planes of ceria. The fact is that the stability of these planes is of the order octahedra {111} > rods {110} > cubes {100} [11].

In recent years, the study of various shapes of cerium oxide has gained importance. As previously mentioned, the structure of nanoctahedrons is the most stable of all. Then there is a cubic structure and finally the rods [13]. The shape plays an important role in lowering the reduction temperature. Thus, it is important to control the shape of the cerium oxide as well as to add metal nanoparticles for the sake of a combination of properties that can have the effect of lowering the reduction temperature to improve catalytic activity. Li et al. studied the improvement of catalytic efficiency using hydrothermal synthesis for CeO_2_ rods [14]. In turn, Yan et al. obtained nanomaterials based on cerium oxide, in which its group was able to demonstrate that by using the principle of coordination chemistry, it is possible to influence the morphology of metal oxides, as well as cerium oxide nanocrystals and to control parameters such as particle size, morphology, surface, texture [15,16]. This proves that a series of active cerium oxide-based catalysts with the desired properties and controlled oxygen vacancies can be obtained.

In 2006, Abanades and Flamanta [17] studied cerium dioxide for the thermochemical splitting of water. In 2010, Chueh et al. [18] used cerium dioxide as a material for solar thermochemical CO_2_ splitting. From that moment on, studies of cerium oxide for thermochemical cycles boosted [19].

The cycle for splitting CO_2_ by cerium dioxide has two stages. The first step is the solar thermal reduction of CeO_2_ in an inert atmosphere and low oxygen vapor pressure to obtain an oxygen-deficient non-stoichiometric cerium oxide. Then, a second step involves the non-solar oxidation of CeO_2-x_ back to CeO_2_, which will take O_2_ directly from H_2_O/CO_2_. The process is visualized in Figure 2. Catalyst performance is determined by the temperature and the partial pressure of oxygen and also depends on the degree of non-stoichiometry. Cerium oxide, while maintaining its fluorite structure, is able to contain large amounts of non-stoichiometric oxygen [20].

Intense research efforts are required to reduce atmospheric CO_2_ levels entailing its conversion in high added value products, such as fuels or chemicals using novel catalysts [22]. CeO_2_-based catalysts doped with metals such as Fe, Pt, Au, Cr, Mo, and Ni were used in CO_2_ splitting. Nickel has been revealed as a good choice due to its low price, relative stability, and good properties for reactions involving CO_2_ [23,24]. However, one of the main drawbacks when using nickel is related to its sintering and formation of undesirable carbon deposits when working under harsh conditions [25]. Both sintering and coke resistance can be improved if nickel is combined with a lanthanoid oxide such as ceria [26]. Therefore, in this work, we focused on the study of the CO_2_ splitting process on nanoshaped CeO_2_ decorated with Pt, Au, Ni, Ru, Cu, and Pd. The influence of the catalyst preparation method (ball milling vs. incipient wetness impregnation) has also been studied.

## 2. Results and Discussion

### 2.1. Nanoshaped Ceria

Appendix A shows the N_2_ adsorption isotherms of the cerium dioxide shapes prepared. The polycrystalline ceria presented an IV-type isotherm, characteristic of mesoporous materials, and an H3 hysteresis loop associated with materials with a regular porous structure and a narrow pore size distribution. The other shapes also showed a type IV isotherm and an H4 hysteresis loop. The type H3 loop is observed with aggregates of plate-like particles giving rise to slit-shaped pores. On the other side, a type H4 loop can be correlated to narrow slit-like pores. Solsona et al. [26] reported similar results. The respective pore size distribution of the samples is presented in Appendix A and indicates that samples are mesoporous with pore diameter higher than 4 nm.

Table 1 summarizes the textural properties of all cerium oxide samples. Surface area (S_BET_) values are in the range of 6.6–63.7 m^2^ g^−1^ and total pore volume (V_tot_) values of 0.06–0.28 cm^3^ g^−1^. The micropore volume of narrow pores (Vmco_2_) (0.3–1.4 nm) ranged from 0.001 to 0.013 cm^3^/g and micropore volume (Vm_N_2__) values were estimated based on N_2_ adsorption ranged from 0.02 to 0.15 cm^3^ g^−1^.

UV-visible absorption spectra of CeO_2_ shapes are shown in Figure 3. All samples have a strong absorption below 400–450 nm. The broad absorption bands located at 250 and 340 nm originate from the charge transfer transition from O^2−^(2p) to Ce^4+^(4f) orbitals in CeO_2_ [27]. The intensity and position of the absorption bands are characteristic of the different shapes of CeO_2_. The band gap values obtained are 2.44, 3.04, 2.60, and 2.69 eV, respectively, for samples CeO_2_ polycrystalline, CeO_2_ rods, CeO_2_ cubes, and CeO_2_ octahedra (Table 2). Similar conclusions have been reported by Patsalas et al. [28] and Filtschew et al. [29].

Figure 4 shows the Raman spectra of the different shapes of cerium oxide. These spectra show an intense peak centered at approximately 465 cm^−1^. This peak is associated with the F_2g_ Raman vibrational mode of the crystalline fluorite structure of ceria, which originated from the tensional vibrations of the oxygen atoms that surround the cerium atoms [30,31]. All cerium dioxide shapes spectra showed characteristic peaks of bulk ceria at 370, 465, 550, 595, 660, and 1170 cm^−1^ [27,32]. According to Filtschew et al. [28] the peaks in the range 370–660 cm^−1^ can be ascribed to second-order Raman peaks. In fact, the former results from a combination of A_1g_, E_g_, and F_2g_ scattering tensors, whereas the latter arises from mixing A_1g_ and E_g_ scattering tensors [32]. The peak around 600 cm^−1^ corresponds to O^2−^ vacancies and the replacement of cerium (IV) atoms by cerium (III) atoms [28,33]. Another small band around 800 cm^−1^ can be attributed to adsorbed peroxide species (O_2_^2−^). The band located around 1170 cm^−1^ was observed in all the spectra of ceria oxide and can be correlated with the Raman mode characteristic of surface superoxide species (O_2_^2−^). A small carbonate peak was observed at about 1060 cm^−1^. The particle size of the CeO_2_ samples was calculated from the Raman line broadening using Equation (3):(3)$$\Gamma \;\left({\mathrm{cm}}^{-1}\right)=10+\frac{124.7}{D}$$
where Γ (cm^−1^) is the full width at half maximum of the Raman active mode peak at 465 cm^−1^ and D is the particle size of the CeO_2_.

The size of the different ceria nanoshapes particles is presented in Table 2.

Figure 5 shows the XRD patterns of the different shapes of cerium dioxide. These XRD patterns were indexed with the JCPDS card no. 81-0792. Eight diffraction peaks were observed at 2θ values, 28.42, 33.13, 47.37, 56.23, 59.04, 69.33, 76.60, and 79.02 corresponding to reflections from the (111), (200), (220), (311), (222), (400), (331), and (420) planes of the cubic crystalline phase of CeO_2_. A good crystallization of all shapes of the cerium dioxide was confirmed with sharp and strong peaks. No additional peaks were observed, indicating the high purity of the synthesized samples. Table 2 presents the average crystallite size for different shapes of cerium dioxide estimated by applying the Scherrer formula.

Figure 6 shows representative SEM images of the different morphologies of cerium dioxide prepared. On measuring the nanoparticles using the Image J software (Version 1.53t), the average size of the particles obtained for each shape was presented in Table 2. The TEM study of all samples was investigated in previous work [33].

H_2_-TPR analysis was performed to understand the reducibility of the various shapes of cerium dioxide (Figure 7). The H_2_-TPR profiles of all samples exhibit a bimodal shape with a wide low temperature peak at 463–555 °C, which is attributed to the characteristic reduction of surface ceria, and a wide high temperature peak at 724−780 °C, which corresponds to the bulk ceria reduction [34,35]. The hydrogen consumption was calculated for each cerium dioxide shape and is presented in Table 3. After H_2_-TPR analysis, the samples were subjected to pulse CO_2_ oxidation to study the oxygen transfer from CO_2_ to reduced ceria through the replenishment of oxygen vacancies created during the reduction treatment with H_2_. Pulse CO_2_ sorption was applied until the cerium oxide was fully saturated. The results for the individual shapes of cerium dioxide are included in Table 3. In all cases, an excellent oxygen exchange was measured, with H_2_ consumption and CO_2_ consumption values virtually identical. The highest reduction/oxidation value was achieved for CeO_2_-rods, and it was about 170 µmol/g_cat_. The lowest value was achieved for CeO_2_-octahedra, with only 20 µmol/g_cat_.

### 2.2. Metal-Decorated Ceria

Generally, polycrystalline ceria nanoparticles usually consist of octahedra or truncated octahedron shapes, which mainly expose the most stable [1 1 1] facets in order to minimize surface energy, whereas nanorods are terminated by [1 1 0] and [1 0 0] planes and nanocubes expose [1 0 0] surfaces. The energy required to form oxygen vacancies on the [1 1 1] surface of CeO_2_ is higher than those on [1 1 0] and [1 0 0] surfaces, so there are more oxygen vacancies on [1 1 0] and [1 0 0] planes [36] After analyzing the reduction/oxidation ability of various ceria nanoshapes, CeO_2_-cubes, and CeO_2_-rods were selected for further studies due to the highest values of oxygen storage-release. Then, they were decorated with 2% wt. of a series of metals: Ru, Pd, Au, Pt, Cu, and Ni. The metal addition positively affects the catalytic performance of bare ceria [37]. As explained in the experimental section, two different methods were used to prepare the ceria samples decorated with metals, incipient wetness impregnation (IWI) and dry ball milling (BM). The redox properties of the obtained catalysts were checked by CO_2_ pulse chemisorption. The samples (50 mg) were first heated at 450 °C in H_2_ (50 mL min^−1^, 10° min^−1^), kept at 450 °C for 30 min, and cooled under Ar flow; then, at room temperature, CO_2_ pulses were injected. Figure 8 shows all of the results.

Nickel turned out to be the most efficient metal for both CeO_2_ cubes and CeO_2_ rods. In addition, the ball milling method for the preparation of the catalysts showed better results in all cases. Therefore, nickel was chosen as a metal and ball milling method to prepare the catalysts for further research. In parallel, an attempt to evaluate the effect of the nickel salt precursor on the redox properties of the catalysts was also investigated (Figure 9). The highest values were obtained for CeO_2_ rods using nickel nitrate. Based on these observations, it was decided to investigate the effect of the amount of metal on the redox capacity of the Ni/CeO_2_-rods catalysts (0.5–10 wt.% Ni).

Appendix A shows the results of the structural and textural characterization of the Ni-CeO_2_ rods samples performed by N_2_ adsorption-desorption isotherms. All isotherms of adsorption-desorption of N_2_ presented in Appendix A showed a type IV isotherm with an H3-shaped hysteresis loop according to the IUPAC classification. This type is typical for powders containing mesopores [38]. Appendix A presents the pore size distribution for all samples. Catalysts exhibited an average pore size of 5–32 nm. An increase of %Ni resulted in a narrowing of the range of pore distribution. The textural parameters of specific surface area (S_BET_), total pore volume (V_tot)_, micropore volume (Vm_N_2__), and pore diameter (D_p_) for the Ni/CeO_2_-rods catalysts are listed in Table 4. The specific surface area and pore volume decreased as the nickel loading increased with respect to pure cerium dioxide. It is likely that pore blocking was caused by nickel accumulation. Conversely, the diameter of the pores was not affected notably by the nickel increase. Peymani et al. [39] and Damyanova et al. [40] reported similar trends. The metal dispersion is calculated using the following equations:(4)$$\mathrm{Metal}\;\mathrm{dispersion}\;\left(\%\right)=\frac{V_{H_{2\;}}\times \;\mathrm{SF}\;\times \;\mathrm{MW}\;}{W_{\mathrm{Ni}}\times 22414}\times 100$$
where V_H_2__ is the volume of the adsorbed H_2_ (mL), SF is the stoichiometry factor, MW is the atomic weight of the Ni (g mol^−1^), and W_Ni_ is the weight of supported Ni on the sample (g). According to XRF analysis, the Ni loadings reached values close to the nominal ones (Table 4).

The XRD patterns of the catalysts are presented in Figure 10. In addition to the characteristic peaks of the fluorite structure of CeO_2,_ peaks at 37°, 44.1°, and 62.6° correspond to NiO (111, 200, and 220 planes, respectively) according to JCPDS file no. 78-0643 [41].

To obtain additional structural information on prepared catalysts, Raman spectroscopy was used, and the corresponding spectra are presented in Figure 11. All the spectra present the main band at 454–463 cm^−1^. This band corresponds to the F_2g_ mode of the fluorite structure of cerium dioxide. The position of the peak shifts from 463 cm^−1^ to 454 cm^−1^ for the catalysts modified with nickel. The reason for this shift can depend on various factors such as oxygen vacancies, phonon confinement, or crystal defects [42,43]. In any case, the peak shift can be related to a strong interaction between the CeO_2_ surface and Ni [42,44]. The peaks observed at 604 and 1177 cm^−1^ correspond to the defect-induced mode (D) and second-order longitudinal optical mode (2LO) bands of ceria, respectively, which become more intense as the amount of Ni increases, pointing out to a strong interaction between CeO_2_ and Ni.

Appendix A shows SEM images showing the morphologies of the prepared catalysts. It can be seen that the structure has not changed with the ball milling method. Rods shape of cerium oxide has been retained for all prepared catalysts.

Figure 12 shows XPS spectra of 2%Ni-CeO_2_-rods. From the spectra, the atomic concentration on the surface is Ce 91.9% and Ni 8.1%, in accordance with the good dispersion of Ni nanoparticles on ceria (Table 4).

Figure 13 shows the H_2_-TPR profiles of the catalysts. It can be seen that the reduction of all catalysts took place in the temperature range of 150–700 °C. As can be seen, three types of reduction peaks (assigned by the symbols α, β, and γ) are observed. The α peak at 150–220 °C was assigned to the surface oxygen located in the defects of the CeO_2_ surface. The peak (β) at 250–280 °C can be attributed mainly to the reduction of NiO weakly interacting with CeO_2_ as a support in the catalysts. The γ peak at 310–350 °C is ascribed to the reduction of NiO strongly interacting with CeO_2_ support [45]. The strong interaction of NiO with CeO_2_ leads to a progressive shift of the γ peak to a higher reduction temperature. Moreover, the area of the reduction peaks in the TPR profile of the catalysts with a higher percentage of nickel was higher than the area detected for the catalysts with a lower nickel content, which was due to the higher amount of hydrogen needed to reduce the catalyst with a higher Ni content, as expected [46].

The quantification of the hydrogen consumption for the Ni-CeO_2_ rod samples is compiled in Table 5, along with the CO_2_ consumption recorded after the H_2_-TPR experiments. In all cases, a good correlation between the oxygen transfer and H_2_ consumption was measured for catalysts with Ni content up to 2%. The best reduction ability was measured for 10%Ni-CeO_2_ rods and it was about 811.6 µmol/g_cat_. The highest reoxidation value was achieved for 2%Ni-CeO_2_-rods, and it was about 412 µmol/g_cat_. As the metal content increases, the reduction capability of the catalyst increases. On the other hand, the ability of reoxidation with the participation of CO_2_ decreases at high Ni loadings, which is a direct consequence of the increasing size of the Ni nanoparticles as the amount of nickel increases. In other words, the oxygen vacancies created at the contact points between Ni and the ceria support is a balance between the loading of Ni and the perimeter available, which is directly related to the size of the Ni particles. How Sarli et al. [37] confirm the strong interaction between metal and CeO_2_ leads to a strong modification of redox properties of ceria, resulting in very active metal-promoted catalysts even at low loads. Table 6 shows the oxygen storage capacity (OSC) for the different catalysts compared with those obtained in this work.

## 3. Preparation and Methods

### 3.1. Preparation of Catalysts

To prepare polycrystalline cerium dioxide (no preferred crystal planes exposed), a solution of 2.0 g of NaOH dissolved in 245 mL of distilled water was prepared using a magnetic stirrer at 300 rpm. A beaker was prepared separately in which 6.07 g of Ce(NO_3_)_3_·6H_2_O were dissolved in 35 mL of distilled water with a magnetic stirrer. Using an ultrasonic atomizer, the cerium salt was added to the aqueous solution of NaOH. The final molarity of the solution was 0.2 M. The resulting solution was then aged for 30 min and introduced in a hydrothermal reactor at 150 °C for 24 h to produce polycrystalline cerium oxide. After cooling, the next step was centrifuging the solution, treating it with ultrasound, and purifying it with distilled water (three times) and ethanol (twice) until the excess NaOH was eliminated from the material. Each centrifugation step was performed at 25 °C, 6000 rpm, and 20 min to achieve complete separation. Then the material was dried at a temperature of approx. 60 °C and crushed in an agate mortar to obtain a fine powder. The last step was calcination at 450 °C for 4 h with a 2° min^−1^ ramp rate.

The cerium dioxide rods and cubes were prepared using the same method but at different concentrations of NaOH in the precursor solution and the temperature of the hydrothermal method. The method of preparing the rods assumes an OH^−^ concentration of 7.875 M. A solution of 6.08 g Ce(NO_3_)_3_·6H_2_O in 35 mL of distilled water was sprayed into a solution of 88.133 g of NaOH in 245 mL of distilled water. Crystallization of the precursor solution in the hydrothermal reactor was carried out at 100 °C for 24 to produce rods. The cubes were prepared with an OH^−^ concentration of the precursor solution of 6 M. A solution of 6.08 g Ce(NO_3_)_3_·6H_2_O in 35 mL distilled water was sprayed into a solution of 67.23 g NaOH in 245 mL distilled water. The precursor solution was treated in a hydrothermal reactor at 180 °C for 24 h.

To obtain cerium dioxide in octahedra shape, 3.26 g of Ce (NO_3_)_3_·6H_2_O were dissolved in 50 mL distilled water using a magnetic stirrer. Then, this solution was spread using an atomizer to the solution of 4.95 mg Na_3_PO_4_·H_2_O with 250 mL distilled water and mixed with a magnetic stirrer for one hour. The next step was to put the solution in the autoclave and heat it at 180 °C for 10 h. After cooling, treatment was the same as for the other shapes.

The catalysts doped with metals were prepared in two different ways, the first of which was the traditionally and commonly used method of incipient wetness impregnation (IWI), which consisted of weighing the appropriate amount of the metal precursor and dissolving it in ethanol, sonicating, and then using an automatic pipette to apply the solution on cerium oxide by spotting. The second method used was ball milling, where a powder mixture placed in the ball mill is subjected to high-energy collision from the balls (BM). For the ball-milled samples, the metal precursor was mixed directly with cerium dioxide in a zirconium oxide vessel (0.5 g of sample and 1 zirconium oxide ball of 15 mm diameter) using a Fritsch Pulverisette 23 (Idar-Oberstein, Germany) mini-mill apparatus. Therefore, after the characterization of different cerium dioxide shapes, the influence of Ru, Pd, Au, Pt, Cu, and Ni (Ru using RuCl_2_, Pd using PdCl_2_, Au using Au(O_2_CCH_3_)_3_, Pt using PtCl_2_, Cu using CuCl_2_, and Ni using NiCl_2_) on the ceria properties was investigated. Catalysts with 2% wt. of metal were prepared by the IWI and BM method. Then, catalysts were based on various shapes of CeO_2_ and various nickel precursors: nickel nitrate (Ni(NO_3_)_2_), nickel chloride (NiCl_2_), and nickel acetate (Ni(CH_3_CO_2_)·4H_2_O) were prepared. After analysis, nickel nitrate was selected for further studies, and catalysts with different wt.% of Ni were prepared using the BM method (15 Hz for 3 min) using CeO_2_ rods at various metal weight concentrations of 0.5, 1, 2, 5, and 10%, respectively. Finally, all prepared samples were calcined at 350 °C with a ramp of 2 °C min^−1^ in the air for 4 h. The prepared Ni/CeO_2_ catalysts were named: 0.5%Ni-CeO_2_ rods, 1%Ni-CeO_2_ rods, 2%Ni-CeO_2_ rods, 5%Ni-CeO_2_ rods, and 10%Ni-CeO_2_ rods according to the nominal wt.% concentration of Ni.

### 3.2. Materials Characterizations

The textural properties of the obtained catalysts were characterized by nitrogen adsorption at −196 °C and CO_2_ adsorption at 0 °C. Pore size distribution (PSD) was obtained from the nitrogen isotherm using nonlinear density functional theory (NLDFT) -assuming slit pores. The micropore volume (V_mCO_2__), up to 1.47 nm, was analyzed via CO_2_ sorption and was also determined by the DFT method. A Quadrasorb Evo instrument (Anton Paar, St. Albans, UK) was used to determine surface area and pore size. The entire study was automated and the results were collected using the QuadraWin program (Versions: 7.1). Prior to the measurements, the catalyst samples were heated at 200 °C for 16 h, with a temperature increment of 1 °C min^−1^, under reduced pressure to remove impurities. Using N_2_ adsorption isotherms, the parameters characterizing the texture of the obtained materials were determined: specific surface area, total pore volume, and micropore volume in the range of 1.4–2 nm. S_BET_ -specific surface area was determined using the Brunauer, Emmett, and Teller equation in the range of partial pressure p/p_0_ = 0.05–0.2. All catalysts were characterized by Raman spectroscopy using a commercial Renishaw inVia Qontor confocal Raman microscope (Gloucestershire, UK). The Raman setup consists of a laser at λ_exc_ = 532 nm with a nominal 100 mW output power. The power of the laser of 1 mW cm^−2^ with 36 accumulations was kept for all samples. To analyze the phase composition of the catalysts, a Bruker D8 Advance diffractometer (Munich, Germany) equipped with a Cu X-ray tube using CuK_α_ radiation (λ = 1.5418 Å, 40 mA, 40 kV) was used. The diffractograms were collected in 2θ range of 20–80° (scan speed = 1 s, step 0.02°). The X Pert HighScore software (version 5.1) was used for the analysis and the Scherrer equation was used for the crystallite size calculation. The ultraviolet-visible (UV–vis) spectroscopy was carried out on a Shimadzu UV3600 UV-vis/NIR apparatus (Oberchoken, Germany). BaSO_4_ was used as a reference standard. The spectra were recorded at room temperature in the air within the range of 300–800 nm. The acquired diffuse reflectance spectra were converted to absorbance through the standard Kubelka-Munk function. The band gap energies (E_g_) of the prepared samples were estimated from the UV-vis spectra by the Tauc method [56,57]. Scanning electron microscopy (Oberchoken, Germany) is used primarily to study the morphology and dimensions of the catalyst particles [58]. All the prepared catalysts were analyzed with a Neon40 Crossbeam™ instrument from Carl Zeiss (Hamburg, Germany). X-ray fluorescence energy dispersion spectrophotometry (EDXRF) was performed on an Epsilon 3 PANalytical B.V. instrument (Almelo, the Netherlands) to determine the content of other elements in the catalysts. To study the reducibility of the catalysts, temperature-programmed reduction (TPR) was used. The TPR method yields quantitative information on the reducibility of the sample and information about the nature of the reducible sites. The TPR experiments were carried out with a Chemstar-TPX instrument equipped (Berlin, Germany) with a thermal conductivity detector (TCD). The samples (50 ± 0.01mg) were first heated from room temperature to 450 °C (10° min^−1^) in a flow of Ar (50 mL min^−1^) and kept at 450 °C for 10 min, then cooled to 50 °C under the same Ar flow rate. The TPR has been carried out from 35 °C to 800 °C (10° min^−1^) under a flow rate of 10 vol. % H_2_ in Ar (with a total flow rate of 50 mL min^−1^) and then the temperature was maintained at 800 °C for 30 min. After H_2_-TPR analysis, the samples were subjected to pulse CO_2_ oxidation to study the oxygen transfer from CO_2_ to reduced ceria through the replenishment of oxygen vacancies created during the reduction treatment with H_2_. Pulse CO_2_ sorption under a flow rate of 10 vol. % CO_2_ in Ar at 25 °C was applied until the cerium oxide was fully saturated with oxygen.

## 4. Conclusions

In this work, four distinct shapes of cerium dioxide (polycrystalline, rods, cubes, and octahedra) were successfully prepared and characterized by N_2_ and CO_2_ adsorption/desorption, UV-VIS, SEM microscopy, Raman spectroscopy, XRD spectroscopy, H_2_- TPR and pulse CO_2_ adsorption. It was concluded that ceria rods exhibit the best CO_2_ splitting capacity. We investigated the effect of IWI and BM method preparation to decorate the ceria nanoshapes with Ru, Pd, Au, Pt, Cu, and Ni. It was found that the samples prepared by BM yielded better CO_2_ splitting results. BM synthesis offers a simple, scalable, and often more sustainable way to synthesize catalytic materials. The key advantages achieved in comparison with wet impregnation are obtaining preparations without the need for additives (e.g., solvents, matrices, or surfactants) and in many cases post-synthetic treatment. This avoids the generation of solvent and/or gaseous wastes and simplifies the entire synthesis protocol, ultimately, simplifying scale-up and, importantly, significantly shortening the catalyst preparation time. In addition, this synthesis strategy provides the materials with extraordinary properties, often impossible to obtain using classical methods. The best results were obtained for a nickel. Furthermore, the type of nickel salt used affected the properties of the resulting catalyst. In our case, the best results were achieved with nickel nitrate. Catalysts with different contents of Ni synthesized using the BM method were analyzed. It was found that the redox properties of surface oxygen vacancy concentration and CO_2_ splitting capacity varied with Ni concentration. 2% Ni-CeO_2_ rods catalyst exhibited the best CO_2_ splitting properties. Our results offer the possibility of using cheap materials prepared by a simple and easily scalable method for CO_2_ splitting. However, long-term and recyclability studies are necessary to have a precise evaluation of their application in practical applications.

## Figures and Tables

**Figure 1 molecules-28-02926-f001:**
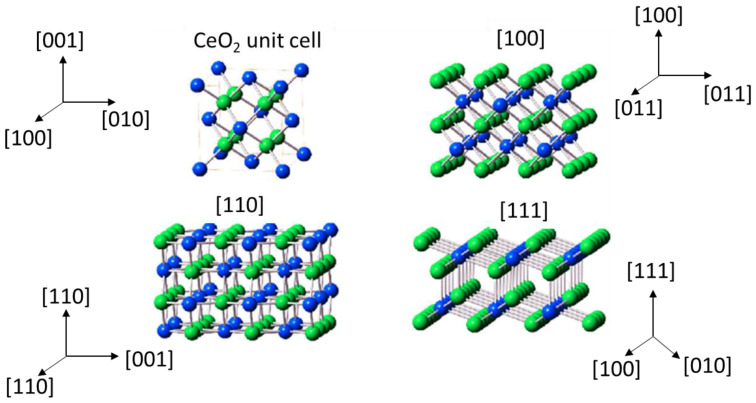
Crystal models of cerium dioxide shapes [12].

**Figure 2 molecules-28-02926-f002:**
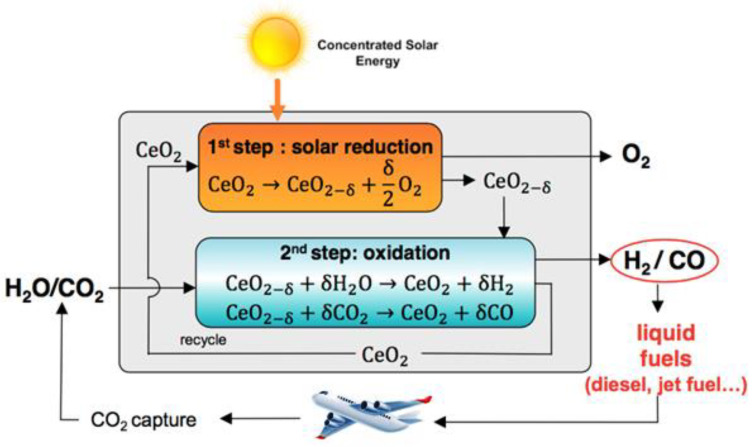
The two-step thermochemical redox process for the splitting of CO_2_ using ceria [21].

**Figure 3 molecules-28-02926-f003:**
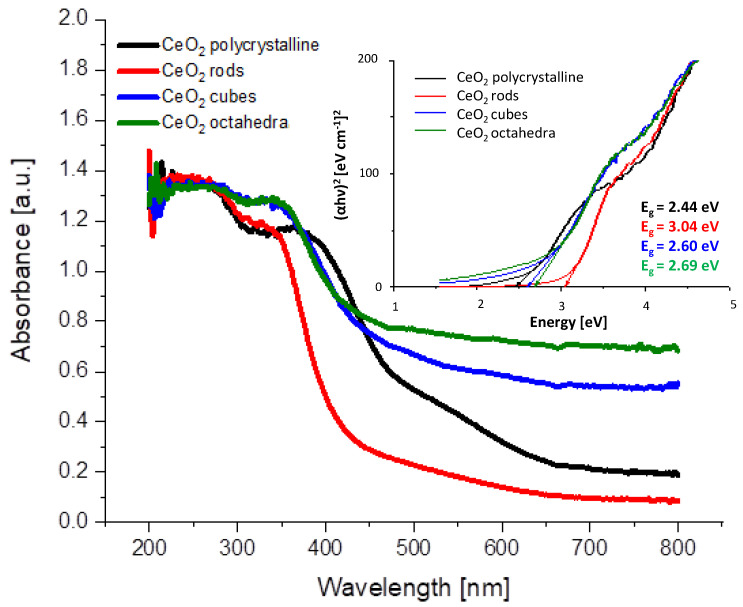
UV-Vis absorption spectra of the prepared cerium dioxide shapes and optical band gap determination from Tauc plots.

**Figure 4 molecules-28-02926-f004:**
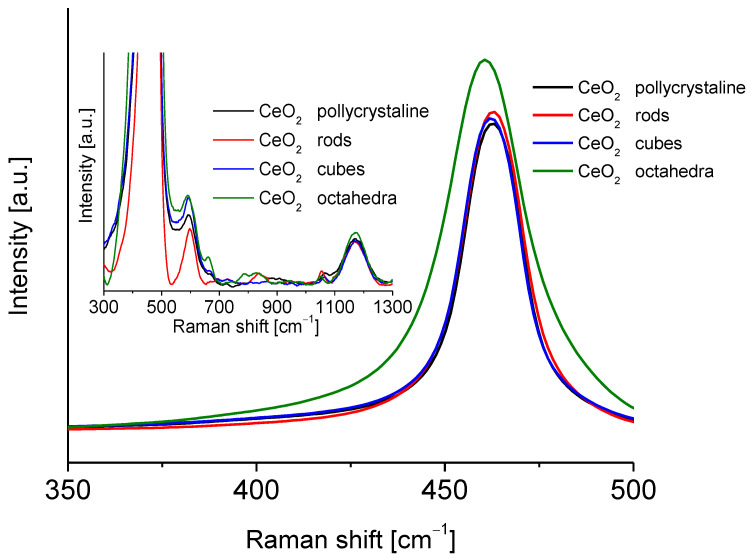
Raman spectra of the cerium dioxide shapes.

**Figure 5 molecules-28-02926-f005:**
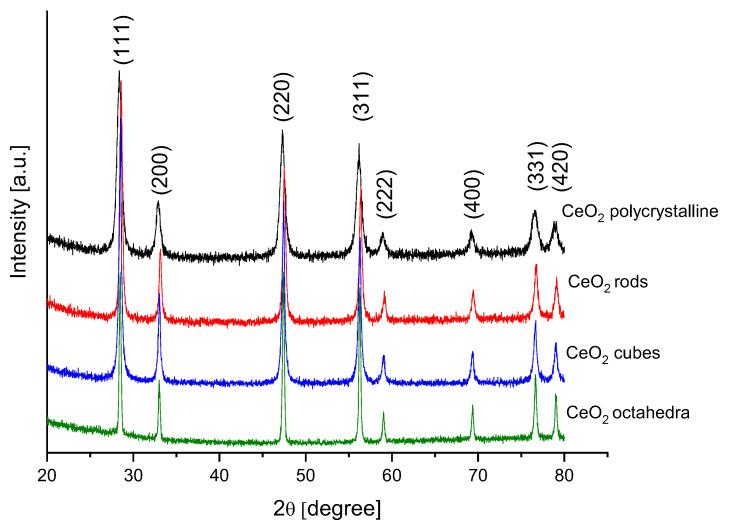
XRD profiles of different cerium dioxide shapes.

**Figure 6 molecules-28-02926-f006:**
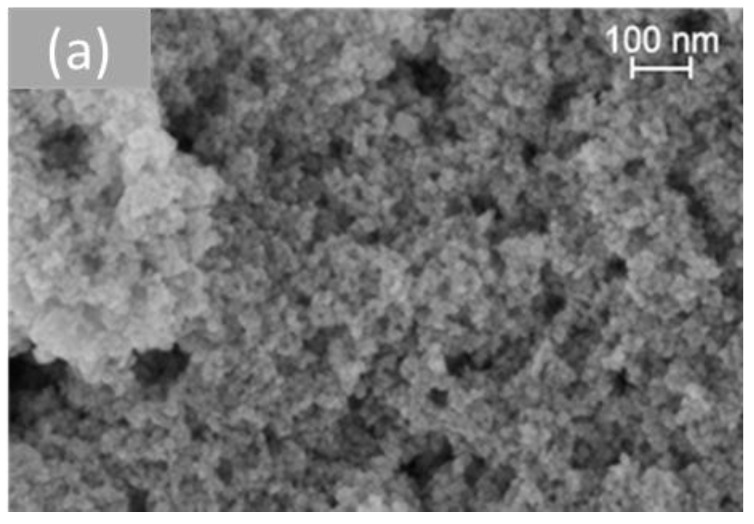
SEM analysis of cerium dioxide shapes: (**a**) CeO_2_ polycrystalline, (**b**) CeO_2_ rods, (**c**) CeO_2_ cubes, and (**d**) CeO_2_ octahedra.

**Figure 7 molecules-28-02926-f007:**
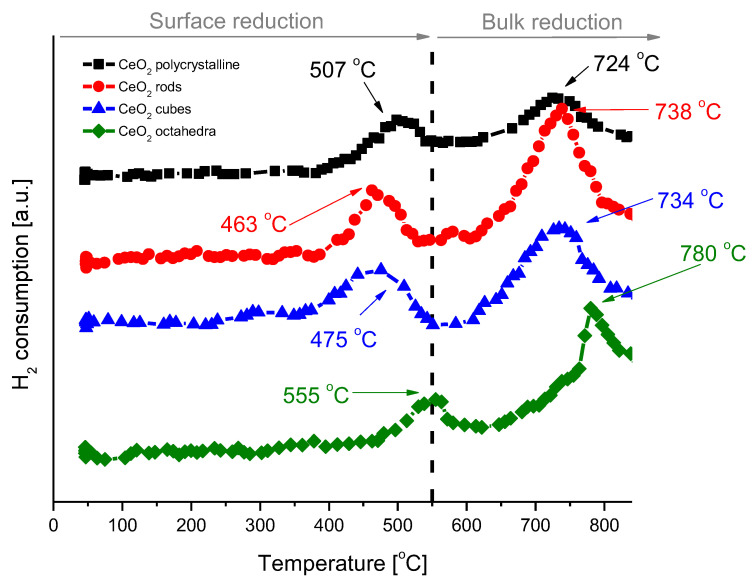
H_2_-TPR profiles of cerium dioxide nanoshapes.

**Figure 8 molecules-28-02926-f008:**
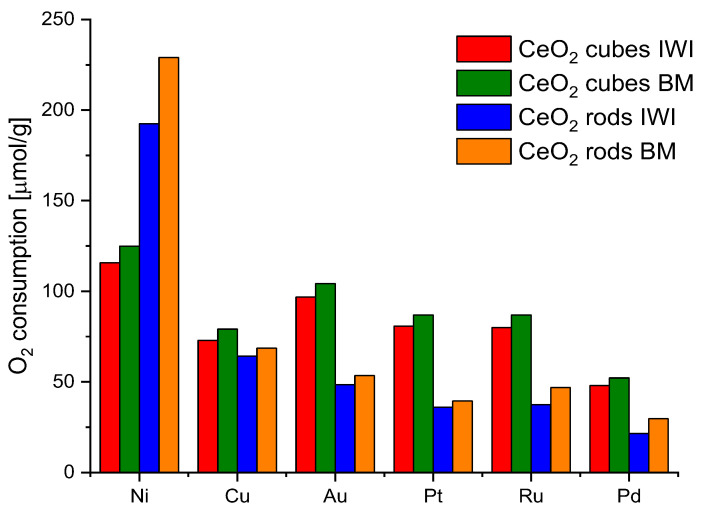
Oxygen consumption from CO_2_ pulse chemisorption of CeO_2_ cubes and CeO_2_ rods decorated with different metals and prepared by different methods.

**Figure 9 molecules-28-02926-f009:**
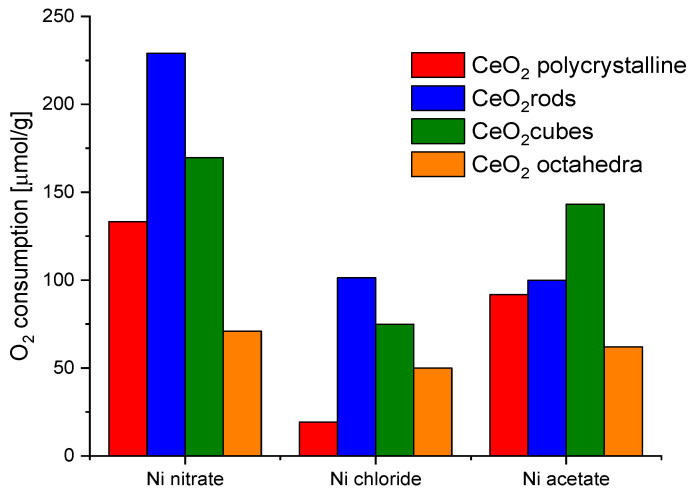
Oxygen consumption from CO_2_ pulse chemisorption of catalysts based on different shapes of CeO_2_ and different nickel salt.

**Figure 10 molecules-28-02926-f010:**
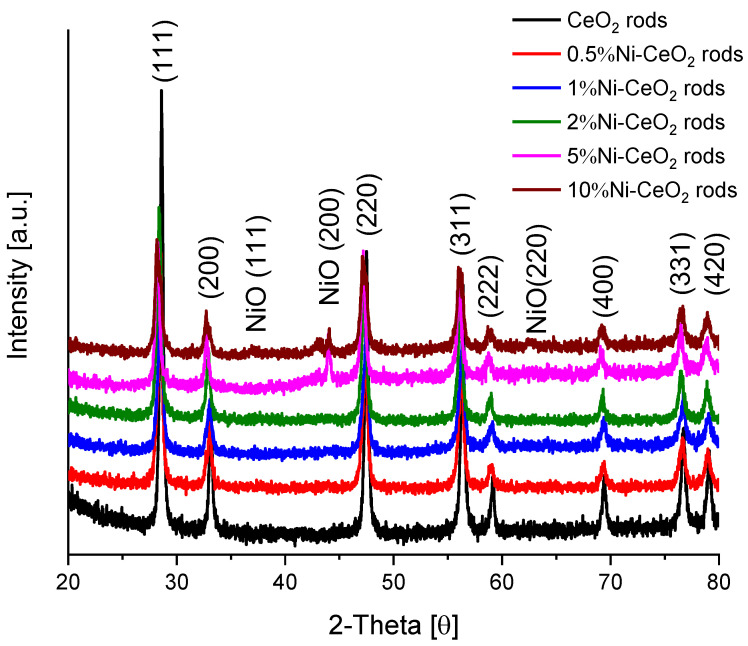
XRD patterns for Ni-CeO_2_-rods catalysts.

**Figure 11 molecules-28-02926-f011:**
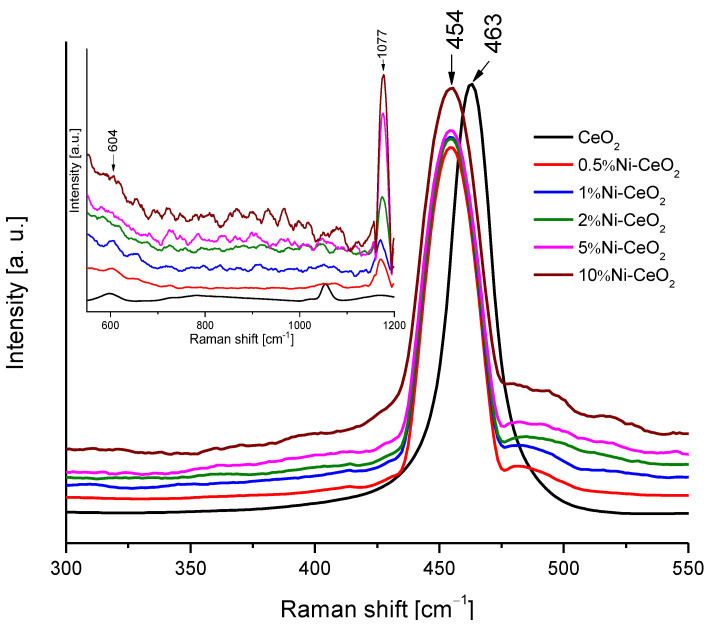
Raman spectra of the Ni-CeO_2_ rods catalysts.

**Figure 12 molecules-28-02926-f012:**
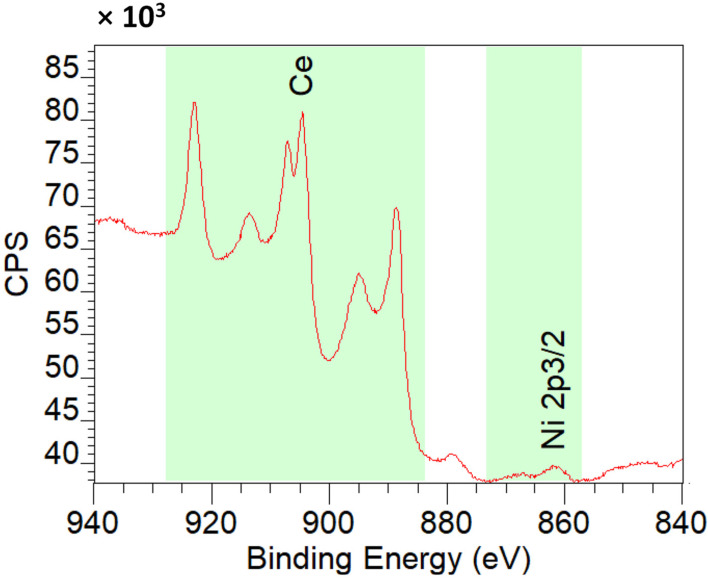
XPS spectra of 2%Ni-CeO_2_-rods.

**Figure 13 molecules-28-02926-f013:**
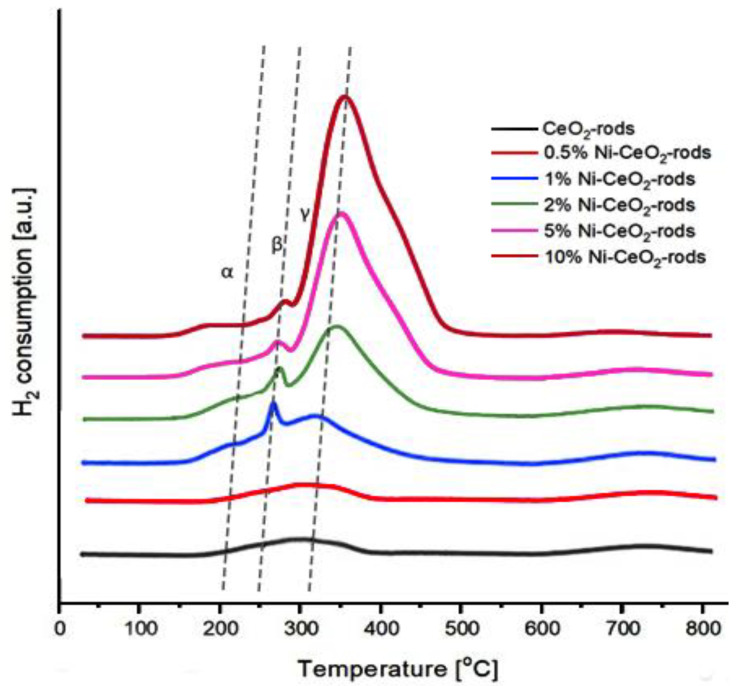
H_2_-TPR profiles of Ni-CeO_2_ rods catalysts.

**Table 1 molecules-28-02926-t001:** Textural properties of cerium oxide shapes.

Catalyst	S_BET_ [m^2^ g^−1^]	Vtot [cm^3^ g^−1^]	Vm_N_2__ [cm^3^ g^−1^]	Vm_CO_2__ [cm^3^ g^−1^]
CeO_2_ polycrystalline	63.7	0.1611	0.145	0.013
CeO_2_ Rods	18.8	0.1060	0.086	0.002
CeO_2_ Cubes	28.9	0.2832	0.129	0.006
CeO_2_ Octahedra	6.6	0.0573	0.019	0.001

**Table 2 molecules-28-02926-t002:** Calculated values of particle size and band gap of cerium dioxide shapes.

Sample	Crystallite Size Calculated Using XRD (nm)	Particle Size Calculated Using RAMAN (nm)	Particle Size Calculated Using SEM (nm)	Band Gap (eV)
CeO_2_ polycrystalline	12.7	13.1	12–13	2.44
CeO_2_ rods	28.2	27.5	26–27	3.04
CeO_2_ cubes	14.3	16.1	15–16	2.60
CeO_2_ octahedra	21.1	20.5	20–21	2.69

**Table 3 molecules-28-02926-t003:** The overall amount of hydrogen consumed in H_2_-TPR and oxygen consumed in CO_2_ pulse chemisorption.

Catalyst	H_2_ Consumption [µmol/g_cat_]	CO_2_ Consumption [µmol/g_cat_]
CeO_2_ polycrystalline	107.1	106.8
CeO_2_ rods	170.1	169.1
CeO_2_ cubes	124.9	124.4
CeO_2_ octahedra	20.3	20.1

**Table 4 molecules-28-02926-t004:** Textural characteristics of Ni-CeO_2_ rods samples and XRF analysis.

Sample	S_BET_ (m^2^ g^−1^)	V_tot_ (cm^3^ g^−1^)	Vm_N_2__ (cm^3^ g^−1^)	D_p_ (nm)	Ni (wt.%) Loading According to XRF	Ni Dispersion (%)
0.5%Ni-CeO_2_ rods	18.7	0.14	0.018	12.4	0.44	4.2
1%Ni-CeO_2_ rods	18.2	0.13	0.015	12.3	1.05	5.4
2%Ni-CeO_2_ rods	17.1	0.12	0.015	12.3	2.09	9.7
5%Ni-CeO_2_ rods	15	0.10	0.010	12.1	5.11	3.9
10%Ni-CeO_2_ rods	10.5	0.05	0.005	11.9	9.86	2.5

**Table 5 molecules-28-02926-t005:** The overall amount of hydrogen consumed in H_2_-TPR and oxygen consumed in CO_2_ pulse chemisorption.

Catalyst	H_2_ Consumption [µmol/g_cat_]	CO_2_ Consumption [µmol/g_cat_]
CeO_2_ rods	170.1	169.1
0.5%-Ni-CeO_2_ rods	179.2	176.8
1%-Ni-CeO_2_ rods	352.4	229.1
2%-Ni-CeO_2_ rods	453.2	412
5%-Ni-CeO_2_ rods	680.1	163.8
10%-Ni-CeO_2_ rods	811.6	105.3

**Table 6 molecules-28-02926-t006:** Oxygen storage capacity (OSC) in μmol CO_2_/g sample for the different catalysts.

Catalyst	OSC [μmol CO_2_/g]	Reference
CeO_2_	10	[47]
CeO_2_/C	255	[45]
Ni-CeO_2_/C	850	[45]
AuCe	103.03	[48]
AuCe50Zr	332.68	[48]
AuCe25Zr	124.88	[48]
CuCe (solegel method)	107	[49]
CuCe (wet impregnation)	476	[50]
Ni/hydroxyapatite	58.80	[51]
Pt/hydroxyapatite	45	[52]
CeO_2_-TiO_2_	66	[53]
Ni/CePr5/Al-1/4	233	[54]
Pt/La(8)/hydroxyapatite	77	[52]
Pd(0.5)/hydroxyapatite	64.2	[55]
2%-Ni-CeO_2_-rods	412	this work

## Data Availability

All data related to this study are presented in this publication.

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
