# Peer review of "Nanoshaped Cerium Oxide with Nickel as a Non-Noble Metal Catalyst for CO2 Thermochemical Reactions"

_molecules, 2023, doi:10.3390/molecules28072926_

Round 1
Reviewer 1 Report
Manuscript Number: molecules-2153718-peer-review-v1
Recommendation: Major revision
Well thought and nicely written manuscript. But the quality could be improved by following the suggestions as follows:
1. As shown in Fig. 2 and discussed in the text the reduction/oxidation state of the CeO2 is important for the reaction. Hence a detailed XPS study is suggested before accepting the manuscript for publication.
2. Mention Redox potential and/or heat of the reaction with the equations 1-2b
3. Blurry figures: 2, 4
4. SEM Images, Fig 7 shows mixed morphology of the CeO2 powder, not CeO2 rods, (c) CeO2 cubes and (d) CeO2 octahedra. In true sense. TEM study is suggested.
5. Authors shown 13 figures in main manuscript and 3 figures in supplementary file. However figures are not organized properly. Proper organizing figures could be better and helpful to enhance quality of the manuscript.
Author Response
We are very grateful to the reviewer for any comments that allowed us to improve the manuscript.
Well thought and nicely written manuscript. But the quality could be improved by following the suggestions as follows:
- As shown in Fig. 2 and discussed in the text the reduction/oxidation state of the CeO2is important for the reaction. Hence a detailed XPS study is suggested before accepting the manuscript for publication.
ANSWER: We fully agree with the referee that the redox state of CeO2 is of paramount importance for the reaction. However, since CeO2 it is well-known that undergoes reduction under vacuum, conventional XPS analysis cannot be properly used for this purpose and only near-ambient pressure XPS (NAPP or APXPS) is suitable for that purpose (see for instance Divins, N., Llorca, J. In situ photoelectron spectroscopy study of ethanol steam reforming over RhPd nanoparticles and RhPd/CeO2. Applied Catalysis A: General, 2016, 518, 60-66). This kind of study requires the use of synchrotron facilities and it is clearly beyond the objectives of this work.
- Mention Redox potential and/or heat of the reaction with the equations 1-2b.
ANSWER: Following the comment of the reviewer, we added information.
- Blurry figures: 2, 4.
ANSWER: Following the comment of the reviewer, we have substituted the initial figures 2 and 4 with better quality ones.
- SEM Images, Fig 7 shows mixed morphology of the CeO2 powder, not CeO2 rods, (c) CeO2 cubes and (d) CeO2 octahedra. In true sense. TEM study is suggested.
ANSWER: We have already reported a detailed TEM study of the samples in a previous work Garcia, X., Soler, L., Casanovas, A., Escudero, C., Llorca, J. X-ray photoelectron and Raman spectroscopy of nanostructured ceria in soot oxidation under operando conditions. Carbon, 2021, 178, 164-180). For the sake of clarity, we have addressed the reader to this publication in the revised version (page 9 line 190).
- Authors shown 13 figures in main manuscript and 3 figures in supplementary file. However figures are not organized properly. Proper organizing figures could be better and helpful to enhance quality of the manuscript.
ANSWER: Following the comment of the reviewer, we have revised the organization of the figures in the revised manuscript.
Reviewer 2 Report
Dear Authors,
As a result of my review, I have the following comments and recommendations:
Manuscript Molecules-2153718 completed by JarosÅ‚aw Serafin and Jordi Llorca refers to the “Nanoshaped Cerium Oxide with Nickel as a Non-noble Metal Catalyst for CO2 Thermochemical Reactions”. In general, nanoparticles considered as major components for nanomaterials are progressing so much that nowadays they are being used almost in every field. Particularly, CeO2 particles, they are now being used for multiple purposes because of their properties and characteristics. In this study, cerium dioxide particles of different formes were prepared and tested for CO2 reduction to CO. The completed results were interpreted through generally accepted and well supported theoretical background. In this manuscript, a set of standard instrumental and wet chemical techniques was used to prepare and characterize the obtained nanoshaped materials. In my opinion, this work could be published in Molecules after minor revision.
1) The paper English is quite good, but main linguistic and scientific revisions aimed to improve its readability.
2) Abstract: some important results (data) should be added such as Hydrogen consumption just to get a clear idea about the whole work.
3) Since all the decorating metals mentioned in the abstract were not used for all prepared shapes. I think it’s better to specify the ones selected and why they were selected?
4) CeO2-cubes and CeO2-rods were selected to be decorated with Ni, why this choice ?
How about the octahedral form which is more stable ?
5) Is there any side effect of CeO2 nanoparticles? If yes, authors should mention it in the main text.
6) Page2, line43 : you should write equation 2a instead of equation1
7) Table1 : check the unit of VmN2
8) How the authors state that the reduction process is or not complete
9) Temperature (450°C) and time (4h) : are these values really sufficient for Calcination? Generally, higher values are needed especially temperature
10)Page 10, line:205: micropore volume (Vmn2) use N2 instead of n2
11) Among other rare earth metals, Why Ce is chosen in this study for samples preparation?
12) The advantages of the method used for Nanoshaped Cerium Oxide with Nickel should be highlighted in comparison to the other coated methods, which would enhance the interest in study of such a kind of synthesis.
13) A table should be included to compare you results with other materials used for the same purpose
14) References [27] and [29] are the same. Also [28] and [32]. Please Check and update in the main text.
15) Most references (>10 years) : reference section should be updated
So after, considering the whole manuscript, all in all, it may be concluded that this paper can be published in Molecules journal after all the scientific and linguistic revisions which are mentioned above are taken seriously by authors.
Author Response
We are very grateful to the reviewer for any comments that allowed us to improve the manuscript.
Manuscript Molecules-2153718 completed by JarosÅ‚aw Serafin and Jordi Llorca refers to the “Nanoshaped Cerium Oxide with Nickel as a Non-noble Metal Catalyst for CO2 Thermochemical Reactions”. In general, nanoparticles considered as major components for nanomaterials are progressing so much that nowadays they are being used almost in every field. Particularly, CeO2 particles, they are now being used for multiple purposes because of their properties and characteristics. In this study, cerium dioxide particles of different formes were prepared and tested for CO2 reduction to CO. The completed results were interpreted through generally accepted and well supported theoretical background. In this manuscript, a set of standard instrumental and wet chemical techniques was used to prepare and characterize the obtained nanoshaped materials. In my opinion, this work could be published in Molecules after minor revision.
1) The paper English is quite good, but main linguistic and scientific revisions aimed to improve its readability.
ANSWER: Following the comment of the reviewer, we have carefully revised the English usage of the manuscript.
2) Abstract: some important results (data) should be added such as Hydrogen consumption just to get a clear idea about the whole work.
ANSWER: According to the reviewer, we have added the hydrogen consumption value of the Ni/CeO2-rods in the abstract and compared the value to the CO2 splitting value.
The highest CO2 splitting value was achieved for 2% Ni/CeO2-rods prepared by ball milling using Ni nitrate (412 µmol/gcat) and the H2 consumption (453.2 µmol/gcat) confirm the good redox ability of this catalyst. (Page 1 line 19).
Page 1 line 17.
3) Since all the decorating metals mentioned in the abstract were not used for all prepared shapes. I think it’s better to specify the ones selected and why they were selected?
ANSWER: To avoid any confusion, we have modified the abstract to better describe the metals used for each shape. (Page 1 line 11).
After an initial analysis based on oxygen consumption from CO2 pulse chemisorption, Ni like a metal and two forms of CeO2 cubes and rods were selected for further research.
4) CeO2-cubes and CeO2-rods were selected to be decorated with Ni, why this choice ? How about the octahedral form which is more stable ?
ANSWER: As discussed in section 3.2, the oxygen consumption from CO2 pulse chemisorption analysis of Ni/CeO2-octahedra and Ni/CeO2-polycrystalline is far below that of Ni/CeO2-rods and Ni/CeO2-cubes (Figure 9). This is the reason why CeO2-cubes and CeO2-rods were selected for further study.
5) Is there any side effect of CeO2 nanoparticles? If yes, authors should mention it in the main text.
ANSWER: No side effects have been observed.
6) Page2, line43 : you should write equation 2a instead of equation1
ANSWER: According to the reviewer suggestion we added.
7) Table1 : check the unit of VmN2
ANSWER: The units have been written correctly in the revised manuscript.
8) How the authors state that the reduction process is or not complete
ANSWER: The reduction was accomplished at 800 °C and the samples were maintained at 800 °C for 30 minutes. This is enough for a complete reduction at this temperature.
9) Temperature (450°C) and time (4h): are these values really sufficient for Calcination? Generally, higher values are needed especially temperature
ANSWER: We have been working with CeO2 for many years and we know that at this temperature no residues remain, as deduced, for instance, from chemical analysis and surface characterization by XPS.
10) Page 10, line:205: micropore volume (Vmn2) use N2 instead of n2
ANSWER: Corrected.
11) Among other rare earth metals, Why Ce is chosen in this study for samples preparation?
ANSWER: CeO2 has been selected for the study since it is the only rare earth with prominent redox behavior. As already explained in the introduction, it is, in fact, the ability of Ce to exchange between Ce(III) and Ce(IV) the reason to choose it for thermochemical reactions.
12) The advantages of the method used for Nanoshaped Cerium Oxide with Nickel should be highlighted in comparison to the other coated methods, which would enhance the interest in study of such a kind of synthesis.
ANSWER: Following the comment of the reviewer, we have added a sentence in the revised manuscript to highlight the advantages of preparing Ni/CeO2 with the ball milling method (page xxx).
BM synthesis offers a simple, scalable and often more sustainable way to synthesize catalytic materials. The key advantages achieved in comparison with wet impregnation is obtaining preparations without the need for additives (e.g. solvents, matrices or surfactants) and in many cases post-synthetic treatment. This avoids the generation of solvent and/or gaseous wastes and simplifies the entire synthesis protocol, ultimately simplifying scale-up and, importantly, significantly shortens the catalyst preparation time. In addition, this synthesis strategy provides the materials with extraordinary properties, often impossible to obtain using classical methods.
13) A table should be included to compare you results with other materials used for the same purpose
ANSWER: According to the reviewer, we have added a Table (new Table 6 in the revised manuscript) to compare the results of this work with others reported in the literature and discussed it in the text (page16).
14) References [27] and [29] are the same. Also [28] and [32]. Please Check and update in the main text.
ANSWER: We are sorry for the inconvenience. We have fixed this issue and revised all the reference list accordingly.
15) Most references (>10 years) : reference section should be updated
ANSWER: We have revised recent literature and added when appropriate (new references).
Reviewer 3 Report
This paper describes an original and interesting work. As such, it has the potential to be published in Molecules. However, I have the following comments that the authors should carefully implement in the revised manuscript prior to publication.
1) Introduction - The connection between the aim of the work and the literature gaps should be better described, thus giving more strength to the reason for this work.
2) Results and discussion - The findings of the following work should also be briefly discussed: Topics in Catalysis, 2021, 64(3-4), pp. 256–269.
3) Results and discussion/Conclusions - The practical impact of the results obtained in this work should be better highlighted.
4) Conclusions - The authors should also give an outlook on future research work.
I’m willing to review the revised manuscript.
Author Response
We are very grateful to the reviewer for any comments that allowed us to improve the manuscript.
This paper describes an original and interesting work. As such, it has the potential to be published in Molecules. However, I have the following comments that the authors should carefully implement in the revised manuscript prior to publication.
1) Introduction - The connection between the aim of the work and the literature gaps should be better described, thus giving more strength to the reason for this work.
ANSWER: Following the comment of the reviewer, we have improved the introduction part. Please check page 3 line 89).
2) Results and discussion - The findings of the following work should also be briefly discussed: Topics in Catalysis, 2021, 64(3-4), pp. 256–269.
ANSWER: We have cited this work and discussed it accordingly in the revised manuscript (pages 11 and 16).
3) Results and discussion/Conclusions - The practical impact of the results obtained in this work should be better highlighted.
ANSWER: We have added a sentence on the practical impact of our results in the conclusions.
Our results offer the possibility of using cheap materials prepared by a simple and easily scalable method for CO2 splitting. However, long-term and recyclability studies are necessary to have a precise evaluation of their application in practical applications.
4) Conclusions - The authors should also give an outlook on future research work.
ANSWER: We have introduced a personal outlook in the conclusions section of the revised manuscript.
Our results offer the possibility of using cheap materials prepared by a simple and easily scalable method to CO2 splitting. However, long-term and recyclability studies are necessary to have a precise evaluation of their application in practical applications.